# IFN-I Score and Rare Genetic Variants in Children with Systemic Lupus Erythematosus

**DOI:** 10.3390/biomedicines12061244

**Published:** 2024-06-03

**Authors:** Rinat K. Raupov, Evgeny N. Suspitsin, Elvira M. Kalashnikova, Lubov S. Sorokina, Tatiana E. Burtseva, Vera M. Argunova, Rimma S. Mulkidzhan, Anastasia V. Tumakova, Mikhail M. Kostik

**Affiliations:** 1Hospital Pediatry Department, Saint-Petersburg State Pediatric Medical University, 194100 Saint Petersburg, Russiaelka_valieva@mail.ru (E.M.K.); lubov.s.sorokina@gmail.com (L.S.S.); 2H. Turner National Medical Research Center for Children’s Orthopedics and Trauma Surgery, Pediatric Rheumatology, 196603 Saint Petersburg, Russia; 3Department of Medical Genetics, Saint-Petersburg State Pediatric Medical University, 194100 Saint Petersburg, Russia; evgeny.suspitsin@gmail.com (E.N.S.); nastenka_0892@bk.ru (A.V.T.); 4Laboratory of Molecular Oncology, N. N. Petrov Institute of Oncology, 197758 Saint Petersburg, Russia; mulkidzhan3@gmail.com; 5Department of Pediatry and Pediatric Surgery, Medical Institute of North-Eastern Federal University, 677007 Yakutsk, Russia; bourtsevat@yandex.ru; 6Yakut Science Center of Complex Medical Problems, Laboratory of Monitoring of the Children Health and Environmental Research, 677018 Yakutsk, Russia; 7Republic Hospital #1–National Center of Medicine, Pediatric Rheumatology, 677010 Yakutsk, Russia; 8Research Laboratory of Autoimmune and Autoinflammatory Diseases, World-Class Research Centre for Personalized Medicine, Almazov National Medical Research Centre, 197341 Saint Petersburg, Russia

**Keywords:** childhood-onset systemic lupus erythematosus, IFN-I score, SLE-associated genes, PTPN22, TREX1

## Abstract

**Introduction:** Interferon I (IFN I) signaling hyperactivation is considered one of the most important pathogenetic mechanisms in systemic lupus erythematosus (SLE). Early manifestation and more severe SLE courses in children suggest a stronger genetic influence in childhood-onset SLE (cSLE). **Aim:** To evaluate IFN-I score and SLE-associated genetic variants in cSLE. **Material and Methods:** 80 patients with cSLE were included in the study. IFN I-score was assessed by real-time PCR quantitation of 5 IFN I-regulated transcripts (IFI44L, IFI44, IFIT3, LY6E, MXA1) in 60 patients. Clinical exome sequencing (CES) was performed in 51 patients. Whole-exome sequencing was performed in 32 patients with negative results of CES. **Results:** 46/60 patients (77%) had elevated IFN-I scores. Leucopenia and skin involvement were associated with over-expression of IFI44 and IFI44L, while hypocomplementemia—with hyperactivation of IFIT3, LY6E, and MX1. No correlation of IFN-I score with disease activity was found. At least one rare genetic variant, potentially associated with SLE, was found in 29 (56.9%) patients. The frequency of any SLE-genetic variants in patients with increased IFN scores was 84%, in patients with normal IFN scores—33%, and in the group whose IFN score was not assessed was 65% (*p* = 0.040). The majority of genetic variants (74%) are functionally related to nucleic acid sensing and IFN-signaling. The highest frequency of genetic variants was observed in Sakha patients (9/14; 64.3%); three and two unrelated patients had identical variants in PTPN22 and TREX1 genes, respectively. **Conclusions:** More than half of patients with childhood-onset SLE have rare variants in SLE-associated genes. The IFN-I score could be considered a tool for the selection of patients for further genetic assessment in whom monogenic lupus is suspected.

## 1. Introduction

Childhood-onset systemic lupus erythematosus (cSLE) is an immune-mediated disease sharing autoimmune and autoinflammatory mechanisms. The disease is characterized by simultaneous or sequential organ and system involvement, with unpredictable course and high levels of morbidity and mortality [1]. The heterogeneity of SLE manifestations might be explained by the variety of contributing pathways. Interferon I (IFN-I) signaling hyperactivation is considered one of the most important pathogenetic mechanisms of SLE. Deoxyribonucleic acid (DNA) and ribonucleic acid (RNA)—containing exogenic and endogenic substrates stimulate plasmacytoid dendritic cells (pDCs). Activated pDCs produce different types of interferons through the hyperexpression of interferon-I regulated genes (IFN-I signature) followed by immune system dysregulation [2,3]. Apoptotic cells, infectious agents, products of neutrophil extracellular traps (NETosis), and autoantibody complexes also activate the IFN I-signaling system.

The majority of the symptoms of SLE (fever, malaise, arthralgia) are associated with hyperproduction of interferons, resembling signs of a viral infection characterized by hyperactivation of the type I IFN pathway [4]. Interferons suppress bone marrow, leading to anemia, neutropenia, lymphopenia, and thrombocytopenia [5]. Patients with active lupus nephritis had high expression of IFN-regulated genes and accumulation of pDCs in glomeruli in the kidney biopsy [6,7]. Previous studies demonstrated the associations between high IFN-I-signature and skin manifestations [8].

IFN-I score was considered as the SLE activity biomarker in different studies, but the results were controversial. Some studies showed a correlation between IFN I score and disease activity scores, another study demonstrated a correlation only with some clinical or laboratory parameters [9,10,11,12].

Children with SLE usually have a more severe disease course than adults, suggesting a more significant genetic background behind pediatric SLE [13]. Currently, up to 100 loci of susceptibility to polygenic, multifactorial SLE and more than 30 genes associated with monogenic lupus and SLE-like phenotypes have been described [14]. Patients with monogenic lupus are characterized by early onset of SLE with severe and unusual clinical manifestations. Monogenic SLE shares clinical manifestations with type I interferonopathies. Patients with chronic atypical neutrophilic dermatosis with lipodystrophy and elevated temperature (CANDLE) syndrome, STING-associated vasculopathy with onset in infancy (SAVI) syndrome, spondyloenchondrodysplasia (SPENCD) syndrome, and Aicardi-Goutières syndromes may have SLE or similar immune-mediated dysregulation in their disease course [15].

Our study aimed to evaluate IFN-I score and SLE-associated genetic variants in cSLE.

## 2. Material and Methods

We included 80 patients under 18 years old with SLE in this prospective study.

The diagnosis of SLE was established according to Systemic Lupus International Collaborating Clinics (SLICC) 2012 criteria [16]. The design of the study is in Figure 1. We evaluated the IFN-I score in 60 patients (main group). The additional group (n = 20) included patients with only the results of a genetic test without IFN-I assessment. The majority of the patients were White/Caucasians (n = 66, 82.5%), and the remaining 14 (17.5%) patients were Sakha ethnicity (the big Asian population in our country).

The patients underwent the following evaluations:Routine clinical and laboratory examinations: complete blood count (CBC), erythrocyte sedimentation rate (ESR), C-reactive protein (CRP), biochemistry, C3- and C4- complements, antinuclear antibodies (ANA), anti-double-stained DNA, the panel of autoantibodies)Assessment of disease activity using the SLEDAI index [17], ECLAM [18]Interferon-I-score (IFN-I score) was measured in 60 patients (main group). IFN I-score was assessed by the reverse transcription polymerase chain reaction, quantitation of five IFN I-regulated transcripts (IFI44L, IFI44, IFIT3, LY6E, MXA1); median relative expression of <2 CU (conventional units) considered as normal, according to previous studies in healthy controls [19]. The IFN I score was re-evaluated during follow-up in 28 patients.Illumina-based clinical exome sequencing (CES) was performed in 51 patients; genes previously shown to be associated with monogenic lupus represented the main interest.Whole-exome sequencing (WES) was performed in 32 patients with negative results of CES.

The indications for genetic testing were: very high disease activity (SLEDAI ≥ 20 points), family history of SLE, the disease onset before seven years, unusual clinical symptoms, high IFN-I score (×10 and more), Sakha ethnicity (Sakha population has the highest frequency in SLE in our country). We selected genetic variants with minor allele frequency (MAF) < 0.5%. The clinical significance of genetic variants was assessed using ACMG (American College of Medical Genetics) criteria and data available in ClinVar and the literature.

### 2.1. Ethics

Written consent was obtained according to the Declaration of Helsinki. The Ethics Committee of Saint Petersburg State Pediatric Medical University (protocol number 1/3 from 11 Januray 2021) approved this prospective study’s protocol. All patients or patients’ representatives (for patients under the age of 15) gave their consent in the patient’s case report forms authorizing the anonymous use of their medical information. All patients were appropriately anonymized.

### 2.2. Statistics

Statistical analysis was performed with the software STATISTICA, version 10.0 (StatSoft Inc., Tulsa, OK, USA). All continuous variables were checked by the Kolmogorov-Smirnov test, and no normal distribution was identified. Continuous variables are presented as median and interquartile ranges (IQRs). Categorical variables are presented as proportions. Missing data were not imputed or not included in the analyses. A comparison of two dependent quantitative variables was carried out using the Wilcoxon test. *p* < 0.05 was considered statistically significant.

## 3. Results

### 3.1. Patient’s Characteristics

The majority of patients were girls (86.7%). The mean age at disease onset was 12.1 (9.2–13.5) years. The median age of inclusion in the study was 14.5 (12.5–16.7) years. A positive family history of immune-mediated diseases had 14 (23%) patients, while 7 (11.5%) patients had first- or second-degree relatives with SLE or similar diseases (overlap syndromes, Sjogren’s syndrome, mixed connective tissue disease, etc). Thirty-nine (65%) patients had high disease activity (SLEDAI >10) at the onset of the disease. All patients received corticosteroids, and 37 (61.7%) of them had treatment with cyclophosphamide or/and rituximab. At the time of inclusion in the study, 38% of patients had an active skin disease, 10% had a fever, 25% had arthritis, 27% had nephritis, 18.7% had central nervous system (CNS) involvement, 56% had hypocomplementemia, 27% had anemia, and 22% had leucopenia. Half of the patients had zero or mild disease activity according to SLEDAI (0–5 points), and 78% of the patients had mild disease activity according to ECLAM (0–5 points).

### 3.2. Assessment of IFN-I Score

#### Initial Assessment

The mean time before the IFN-I score assessment was 18.0 (5.5; 44.0) months. The majority of the patients (46/60; 76.7%) had elevated IFN-I scores. The patients with high IFN-score had skin involvement (43.5% vs. 21.4%), fever (13.0% vs. 0%), anemia (21.7% vs. 5.0), neutropenia (11.7% vs. 0%), hypocomplementemia (62.5% vs. 30.0%) more often compared patients with normal IFN-I score (differences are non-significant), and required more aggressive treatment with rituximab and/or cyclophosphamide (67.4% vs. 42.9%). There were no differences in activity scores (SLEDAI, ECLAM) between these two groups. The detailed comparison is in Table 1.

We found positive correlations between IFN-I score, IFN-regulated genes, and SLE parameters (Appendix A). Patients with leucopenia had higher levels of IFI44 (27.9 CU vs. 7.1 CU, *p* = 0.033), IFI44L (5.8 CU vs. 2.2 CU, *p* = 0.013), MX1 (11.6 CU vs. 3.8 CU, *p* = 0.048), compared with patients without leucopenia. Patients with low complement had higher levels of IFN-I score (10.2 CU vs. 3.6 CU, *p* = 0.025), IFIT3 (8.7 CU vs. 3.5 CU, *p* = 0.019), LY6E (6.5 CU vs. 1.3 CU, *p* = 0.034), MX1 (7.1 CU vs. 2.1 CU, *p* = 0.029). Patients with skin involvement had higher levels of IFN-I score (12.7 CU vs. 4.6 CU, *p* = 0.052), IFI44 (24.6 CU vs. 7.1 CU, *p* = 0.048), IFI44L (36.6 CU vs. 10.4 CU, *p* = 0.023).

### 3.3. Follow-Up IFN-I Score Assessment

We assessed the IFN-I score repeatedly in 28 patients through 8 (5; 10) months. Twenty-one (75%) patients had increased second IFN-I score measurement. One patient with a normal IFN-I score at the baseline had a decreased SLEDAI score. IFN-I score decreased in thirteen patients during the study, but its level continued to be high. In the patients who demonstrated a decreasing IFN-1 score, the SLEDAI score was reduced in seven patients. SLEDAI score increased in three patients, while 3/13 patients had an unchanged SLEDAI score. In six of seven patients with elevated IFN-I scores over time, SLEDAI scores decreased, and in the remaining patients, SLEDAI scores increased. The IFN-I score and SLEDAI dynamics during the study are in Table 2. There were no significant correlations between changes in the IFN-I score and disease activity.

### 3.4. Genetic Testing

Twenty-nine (56.9%) patients had at least one rare genetic variant (MAF < 0.5%) potentially associated with SLE. In total, we found 31 different genetic variants. They were pathogenic or likely pathogenic variants (39%), variants of unknown significance (VUS, 44%), benign (17%), or likely benign (17%). Data are in Table 3.

Eight patients (15.7%) had relatives with SLE, and four of them had the same genetic variants as their relatives with SLE. The frequency of any SLE-associated genetic variants depended on the IFN score. Patients with increased IFN scores had these variants frequently (84%) compared to individuals with normal IFN scores (33%) and patients whose IFN score was not assessed—65% (*p* = 0.040).

We found twelve different genetic variants associated with SLE in 9/14 (64.3%) Sakha patients. Three Sakha patients had identical rare VUS c.1127C>T in the *PTPN22* gene, and two patients had identical likely pathogenic variant c.-26-1G>A in *the TREX1* gene.

The majority of genetic variants (74%) were responsible for nucleic acid sensing and IFN-I signaling. Other variants were associated with apoptotic and immune complexes clearance (8.7%), NF-kappa-B (8.7%), and immune cell signaling (8.7%).

## 4. Discussion

The majority of our patients had activation of the IFN-1 signaling pathway. Patients with increased IFN-1 scores had variants in SLE-associated genes associated with nucleic acid sensing and IFN-1 signaling. A finding of SLE biomarkers is a challenging problem [19,20,21].

According to previous studies, about 50–70% of adults and 90% of children with SLE had characteristic IFN-stimulated gene expression profiles [2]. Our study confirmed the hyper-expression of IFN-I-regulated genes in the majority of our SLE patients: 77% of them had an elevated IFN-I score. Hyperactivation of the IFN-I signaling cascade plays a crucial role in skin manifestations in SLE. Several studies demonstrated a similar type of rash in SLE and monogenic interferonopathies, high IFN-regulated gene expression in a skin biopsy, and efficacy of inhibitors of the IFN-I signaling system [8,22,23].

The new treatment approach in SLE—the monoclonal antibodies to the type I interferon receptor blocking the IFN-1-mediated signaling pathway was recently approved in adults. It showed efficacy in moderated lupus, lupus-nephritis, and skin disease [24,25,26,27]. Litifilimab (BIIB 059), a humanized monoclonal antibody, binding blood dendritic cells antigen 2 (BDCA2), a protein uniquely expressed on plasmacytoid dendritic cells, indirectly inhibits IFN-1 signaling pathway showed efficacy in skin manifestations of SLE [28].

Previous studies showed a significant positive correlation between peripheral blood IFN-I score and clinical manifestations such as lupus nephritis and SLE activity scores [29,30]. However, subsequent studies questioned the use of the IFN-I score as a disease activity biomarker since the IFN-I score remained stable over time in most of the patients despite changes in the disease activity [31]. Northcott and colleagues evaluated the IFN-I score in 205 patients with SLE: 63% of the patients had high IFN-I scores at baseline, and 87.3% of the patients demonstrated stable IFN-I score levels over time [12]. A high IFN-I score correlated with increased levels of anti-dsDNA and low complement in 243 adult SLE patients. However, the reduction of serological activity markers has not been accompanied by the reduction of IFN-I score levels in SLE patients, similar to our results [9]. A.Tesser and colleagues selected a group of children with SLE with high IFN-I score and normal complement levels. IFN-I signature did not correlate with disease activity scores in the study [10].

The analysis of IFN-I score (LY6E, OAS1, OASL, MX1 и ISG15) in 48 patients with SLE showed the associations with SELENA-SLEDAI score, and LY6E expression, which was high in patients with active lupus nephritis [11]. Lambers and colleagues found similar results: IFN-I score correlated with low complement, skin disease, alopecia, and SLEDAI score [32].

The results of our study did not confirm the links between IFN-I score and disease activity, except for the association between low complement and IFN-I score, LY6E, MX1, and IFIT-3 hyper-expression.

Wahadat MJ et al. showed high IFN-I scores in 57% of children with SLE. Increased IFN I score was associated with hyper-expression of TLR7, and RNA and DNA binding receptors genes, contributing to IFN-I activation via TBK1 signaling [33].

The analysis of 1832 candidate genes in 958 SLE patients and 1026 healthy individuals identified two main pathways associated with T-lymphocyte differentiation and innate immunity (HLA and interferons) disturbances. Organ damage in SLE patients was associated with the T- or B-cell receptor signaling pathways [34]. In our group, the majority (67%) of detected variants were responsible for components of the IFN-I signaling system, and all these patients had high IFN-I scores. The IFN-I score could, therefore, be used as a pre-genetic tool for the selection of the patients for the following genetic studies.

Early disease onset, family history of autoimmune/immune-mediated diseases, syndromal SLE, and severe disease course were criteria for WES in Chinese children with SLE. In 52/281 patients who satisfied one of these criteria, five rare causative variants were found in SLC7A7, NRAS, TNFAIP3, PIK3CD, and IDS genes [35]. In our group, 4/8 (50%) patients with positive family history had genetic variants identical to their relatives. Two patients were monozygotic twin brothers with the *RNASEL p.K627R* variant, but only one of them developed full-blown SLE, and the second twin had identical typical SLE activity (high ANA, antidsDNA, and antiphospholipid antibodies) and developed panniculitis [36].

Patients of Sakha ethnicity were of particular interest in our study due to genetic homogeneity and high prevalence of hereditary and immune-mediated diseases (SLE, ankylosing spondylitis, Behсet’s disease, IgA vasculitis, Takayasu arteritis) [37]. They had a higher prevalence of rare genetic variants compared to Caucasians. We found identical rare c.1127C>T genetic variant (VUS) in the *PTPN22* gene and c.-26-1G>A pathogenic genetic variant in *TREX1*gene previously not reported in ClinVar.

PTPN22 (Protein tyrosine phosphatase non-receptor type 22) protein is a negative regulator of T-cell receptor signaling. Polymorphisms in the PTPN22 gene were described in the broad spectrum of immune-mediated diseases, including SLE [38,39]. *PTPN22* rs1310182 genetic variant is considered a genetic marker for susceptibility to pediatric SLE [40].

It is known that *TREX1* genetic variants may lead to IFN-I signaling hyper-activation. Compound heterozygous and homozygous missense and nonsense genetic variants in the *TREX1* gene were detected in 25% of cases of Aicardi-Goutières syndrome, and 60% of patients with *TREX1* variants had at least one of the following signs: antinuclear antibodies, antibodies to dsDNA, thrombocytopenia, leukopenia, skin rash, oral ulcers, arthritis. Most of the patients with *TREX1* variants had increased IFN-I scores [41]. The frequency of heterozygous missense variants of the *TREX1* gene in SLE was up to 2% [42].

The first patient with *TREX1 c.-26-1G>A* variant was a 13-year-old boy with SLE with psychiatric problems, livedoid rash, aphthous stomatitis, pneumonitis, thrombocytopenia, and lymphopenia. The second patient was a girl with malar rash, polyarthritis, oral ulcers, anemia, and leucopenia. This variant previously was only once reported in Iranian female with Aicardi-Goutières syndrome type I [43].

TREX 1 and PTPN22 variants were therefore over-presented in our small group of Sakha children with SLE. However, further studies using a wider sample of patients are required.

### Study Limitations

Our study has several limitations related to the heterogeneous studied population. The assessment of IFN-I score was performed at different time points, and in some patients, previous treatment might influence the level of IFN-I score as well as the following treatment. Different treatment protocols and assessments at different time points did not allow for assessing the role of IFN-I score more thoroughly and finding a more precise association with SLE activity. The impossibility of performing WES in all studied patients might cause some other genetic variants to be missed, as well as the absence of a trio genetic study and functional testing for some genes.

## 5. Conclusions

IFN-I score might be considered as a tool for the selection of patients for further genetic studies, for whom monogenic lupus is suspected. More than half of patients with childhood-onset SLE have rare variants in SLE-associated genes. TREX 1 and PTPN22 variants were over-presented in the group of Sakha children with SLE. However, further studies are required.

## Figures and Tables

**Figure 1 biomedicines-12-01244-f001:**
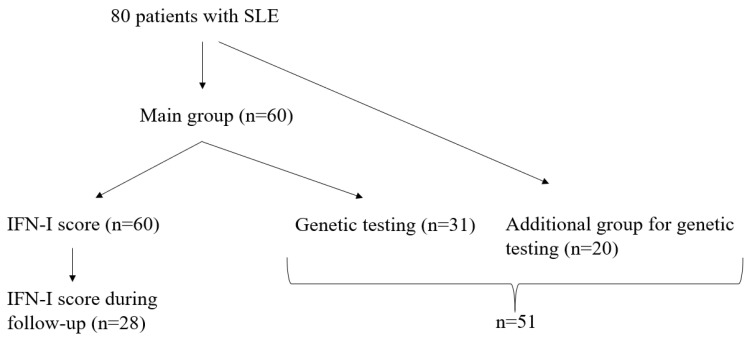
The study design.

**Table 1 biomedicines-12-01244-t001:** Clinical and laboratory features in patients with high and normal IFN-I scores.

Parameter	All Patients, n = 60	High IFN-I Score, n= 46	Normal IFN-I Score, n = 14	*p*-Value
Girls, n (%)	52 (87)	47 (78)	13 (93)	0.105
The age of inclusion, years	14.5 (12.5;16.7)	14.5 (12.6;16.6)	14.8 (10.6;16.7)	0.937
The age of onset, years	12.1 (9.2;13.5)	12.2 (10.1;14.6)	11.4 (7.7;12.8)	0.990
The time from onset to IFN-I score evaluation, months	18.0 (5.5;44.0)	17.5 (5.0;43.0)	26.0 (7.0;45.0)	0.736
Skin involvement, n (%)	23 (38)	20 (44)	3 (21)	0.137
Mucosa involvement, n (%)	6 (10)	5 (11)	1 (7)	0.684
Alopecia, n (%)	7 (12)	4 (9)	3 (21)	0.194
Arthritis, n (%)	15 (25)	11 (24)	4 (29)	0.724
Nephritis, n (%)	16 (27)	13 (28)	3 (21)	0.613
CNS-involvement, n (%)	11 (19)	10 (22)	1 (7)	0.454
Hepatomegaly, n (%)	4 (7)	4 (9)	0 (0)	0.253
Lymphadenopathy, n (%)	2 (3)	2 (4)	0 (0)	0.427
Lung involvement, n (%)	2 (3)	3 (7)	0 (0)	0.327
Raynaud’s phenomenon, n (%)	5 (8)	5 (11)	0 (0)	0.198
Fever, n (%)	6 (10)	6 (13)	0 (0)	0.154
Hypocomplementemia, n (%)	28/50 (56)	25/40 (63)	3/10 (30)	0.064
Anemia, n (%)	16 (27)	13 (22)	3 (5)	0.613
Leucopenia, n (%)	13 (22)	11 (183)	2 (3)	0.444
Neutropenia, n (%)	7 (12)	7 (12)	0 (0)	0.120
Lymphopenia, n (%)	19 (32)	15 (25)	4 (7)	0.776
Thrombocytopenia, n (%)	3 (5)	2 (3)	1 (2)	0.674
SLEDAI, points	5 (2; 11)	8 (2; 12)	2.5 (0; 8)	0.135
SLEDAI (0 points), n (%) SLEDAI (1–5 points), n (%) SLEDAI (6–10 points), n (%) SLEDAI (11–19 points), n (%) SLEDAI (≥20 points), n (%)	11 (18) 20 (33) 9 (15) 13 (22) 7 (12)	7 (15) 14 (30) 8 (17) 11 (24) 6 (13)	4 (29) 6 (43) 1 (7) 2 (14) 1 (7)	0.546
ECLAM, points	2 (0.7; 4)	2 (1; 4)	1 (0; 2)	0.080
ECLAM (0 points), n (%) ECLAM (1–5 points), n (%) ECLAM (6–9 points), n (%) ECLAM (≥10 points), n (%)	2 (3) 47 (78) 10 (17) 1 (2)	2 (4) 34 (74) 9 (20) 1 (2)	0 (0) 13 (93) 1 (7) 0 (0)	0.494
Treatment				
Hydroxychloroquine, n (%)	57 (95)	43 (93.5)	14 (100)	1.0
Prednisone, n (%)	60 (100)	46 (100)	14 (100)	1.0
Methotrexate, n (%)	9 (15.0)	8 (17.4)	1 (7.1)	0.671
Mycophenolate mofetil, n (%)	31 (51.7)	24 (52.2)	7 (50.0)	1.0
Azathioprine, n (%)	3 (5.0)	2 (4.3)	1 (7.1)	0.556
Cyclophosphamide, n (%)	18 (30.0)	14 (30.4)	4 (28.6)	1.0
Rituximab, n (%)	34 (56.7)	28 (60.9)	6 (42.9)	0.356
Belimumab, n (%)	6 (10.0)	5 (10.9)	1 (7.1)	1.0
IVIG, n (%)	24 (40.0)	19 (41.3)	5 (35.7)	0.765

Abbreviations (in alphabetical order): CNS—central nervous system; ECLAM—European Consensus Lupus Activity Measurement; IFN I—interferon type I; IVIG—Intravenous Immunoglobulin; SLEDAI—systemic Lupus Erythematosus Disease Activity Index.

**Table 2 biomedicines-12-01244-t002:** IFN-I score and disease activity during the follow-up.

First IFN-Score, n	Second IFN-Score, n	SLEDAI Change, n
Normal—5 patients	Normal—4 patients	No change—4 patients
	Increased—1 patient	Decrease—1 patient
Elevated—23 patients	Decreased to normal ranges—3 patients	Decrease—2 patients No change—1 patient
	Decreased over time, but still high—13 patients	Decrease—7 patients No change—4 patients Increase—3 patients
	Increased—7 patients	Decrease—6 patients Increase—1 patient

**Table 3 biomedicines-12-01244-t003:** The distribution of genetic variants associated with SLE.

Pathogenic/Likely Pathogenic	VUS	Non-Pathogenic Benign/Likely Non-Pathogenic Benign
*DDX58* NM_014314 c.2590_2591insTTCT (p.C864Ffs*9)	*RNASEL* NM_021133 c.1880A>G (p.K627R) (n = 2)	*IFIH1* NM_022168 c.1879G>T (p.E627X)
*IFI35* NM_005533.5 c.541_542del (p.Val181fs*)	*IFNAR2* NM_001289125 c.C1451T (p.P484L)	*TREX2* NM_080701.3 c.176A>G (p.K59R)
*NLRP14* NM_176822 c.606delG (p.G203Afs*21)	*RNASEH2B* NM_024570 c.794T>C (p.F265S)	*DNASE1L3* NM_004944 c.288C>A (p.N96K)
*TMEM173* NM_198282 c.1013delA(p.K338Rfs*8)	*IRF5* NM_001098629 c.389C>T (p.S130F)	*TLR1* NM_003263 c.1009C>T (p.R337C)
*IFI44* NM_006417.5 c.163-2A>G	*TLR3* NM_003265 c.1311C>A (p.D437E)	*TLR1* NM_003263 c.893C>T (p.S298F)
*SAMHD1* NM_015474 c.428G>A (p.R143H)	*TLR8* NM_138636.5 c.1172T>C (p.M391T)	*DNASE1* NM_005223 c.460C>G (p.P154A)
*RAB44* NM_001257357.2 c.214A>T (p.Lys72*)	*RNASEH2B* NM_024570 c.295C>T (p.H99Y)	*CR1* NM_000573 c.313C>T (p.R105C)
*RNASEH2C* NM_032193 c.268_270del (p.K90del) homozygous	*CR1* NM_000573 c.1716G>T (p.Q572H) (n = 2)	*CR2* NM_001006658.3 c.641G>A (p.Arg214His)
*TREX1* NM_033629.6: c.-26-1G>A (n = 2)	*TNFAIP3* NM_001270507 c.2364G>A (p.M788I)	
*C1QA* NM_015991 c.334C>T (p.Q112X) homozygous	*CASP8* NM_001372051.1 c.892A>G (p.I298V)	
*PTPN11* NM_002834 c.226G>A (p.Glu76Lys)	*ADAR* NM_001111 c.643G>A (p.G215S)	
	*PTPN22* NM_015967 c.1127C>T (p.T376I) (n = 3)	

## Data Availability

Data are available on request from the authors.

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
