# Peer review of "IFN-I Score and Rare Genetic Variants in Children with Systemic Lupus Erythematosus"

_biomedicines, 2024, doi:10.3390/biomedicines12061244_

Round 1

Reviewer 1 Report

Comments and Suggestions for Authors Interesting and well developed paper. But there are some criticisms to evaluate. Many abbreviations are used. Insert a legend at the end or insert the full name the first time in the text. Check.
In table 1 insert the treatment evaluation (rituximab and cyclophospamide). Paragraph 3.3 is not very clear. Why 8 months were chosen? Insert observations on the results described in table 2.
Inserted like this it does not blend well with the rest of the paper.

Author Response

Dear Reviewer!

Thank you so much for your positive evaluations of our manuscript. Our answers (A) on your queries (Q) are below and highlighted by color in the manuscript.

Reviewer 1

Q1. Interesting and well developed paper. But there are some criticisms to evaluate. Many abbreviations are used. Insert a legend at the end or insert the full name the first time in the text.

A1. Dear Reviewer! The full name added the first time in the text abbreviations mentioned (for frequent abbreviations), also I added the full name in the bottom of the table and whole list of abbreviations added at the end of the manuscript. All full names provided in the alphabetical order.

Check.
Q2. In table 1 insert the treatment evaluation (rituximab and cyclophospamide).

A2. Dear Reviewer, added.

Q3. Paragraph 3.3 is not very clear. Why 8 months were chosen? Insert observations on the results described in table 2. Inserted like this it does not blend well with the rest of the paper.  

A3. We tried to make the second assessment in 6 months (repeated courses of rituximab of finishing of cyclophosphamide induction course), but our patients live in very big distance, especially from Sakha Republic (6000 km), so it was very difficult to keep time intervals. Due to centralization of the specialized healthcare some patients were admitted in our hospital and I had several business trips in Sakha Republic. The median time was 8 months. This was not a specific time interval planned initially.

The explanation of the changes, mentioned in Table 2 added in the text above the table.

Dear Reviewer!

I hope the manuscript became better after your suggestions and recommendations.

On behalf of the Authors

Mikhail Kostik, MD, PhD, Professor

Reviewer 2 Report

Comments and Suggestions for Authors

The authors present an interesting IFN score and genetic variants in childhood SLE. The article is well structured. Here you are my comments.

1.        Please define abbreviations before using them in the main text.

2.        Please revise and correct some typos all over the article (e.g. page 1 line 51 “viral infection there….”, page 2 line 76 “The additional group (n=20) in patients with only genetic test results…”, page 3 line 122 “high disease activity … had 39 patients” and several others).

3.        Results: please uniform presentation in past tense. 

4.        Results: when you report that patients with high IFN-score more often had skin involvemente, neutropenia, etc you should specify that no difference reached significance.

5.        Genetic testing is supposed to be part of results section not a separate one. 

6.        How many patient in your cohort got whole exome sequencing?

Comments on the Quality of English Language

Several typos, but flows quite well. 

Author Response

Reviewer 2

The authors present an interesting IFN score and genetic variants in childhood SLE. The article is well structured. Here you are my comments.

Dear Reviewer!

Thank you so much for your positive evaluations of our manuscript. Our answers (A) on your queries (Q) are below and highlighted by color in the manuscript.

Q1. Please define abbreviations before using them in the main text.

A1. Dear Reviewer! The full name added the first time in the text abbreviations mentioned (for frequent abbreviations), also I added the full name in the bottom of the table and whole list of abbreviations added at the end of the manuscript. All full names provided in the alphabetical order.

Q2.        Please revise and correct some typos all over the article (e.g. page 1 line 51 “viral infection there….”, page 2 line 76 “The additional group (n=20) in patients with only genetic test results…”, page 3 line 122 “high disease activity … had 39 patients” and several others).

A2. Dear Reviewer! The typos were fixed.

Q3. Results: please uniform presentation in past tense. 

A3. Dear Reviewer! Done.

Q4. Results: when you report that patients with high IFN-score more often had skin involvemente, neutropenia, etc you should specify that no difference reached significance.

A4. Dear Reviewer! The information about the absent of the significance was added in the text.

Q5. Genetic testing is supposed to be part of results section not a separate one. 

A5. Dear Reviewer, done. The following re-numeration was also done

Q6. How many patient in your cohort got whole exome sequencing?

A6. Dear Reviewer, the WES was done in 32 patients.

Comments on the Quality of English Language Several typos, but flows quite well. 

Dear Reviewer!

I hope the manuscript became better after your suggestions and recommendations.

On behalf of the Authors

Mikhail Kostik, MD, PhD, Professor

Reviewer 3 Report

Comments and Suggestions for Authors

Dear Sirs,

this is an interesting work, that merits publication. However, there are some points that need to be improved. The English language needs substantial improving. Regarding the study, the authors should acknowledge in a table the drugs that the patients had been taking and they should state whether or not they have been administered the monoclonal antibody against INF-1.  In addition, the discussion should be more focused upon their findings and their potential explanations regarding their findings. Finally, apart from improving the discussion, more recent referencing from 2023 and 2024 should be added in the discussion and the reference section. Perhaps the authors might want to comment on the monoclonal antibody against INF-1 as well, which has very recently been released.

Comments on the Quality of English Language

POOR QUALITY. NEEDS SUBSTANTIAL IMPROVEMENT.

Author Response

Reviewer 3

Dear Reviewer!

Thank you so much for your positive evaluations of our manuscript. Our answers (A) on your queries (Q) are below and highlighted by color in the manuscript.

Dear Sirs,

this is an interesting work, that merits publication. However, there are some points that need to be improved. The English language needs substantial improving. Regarding the study, the authors should acknowledge in a table the drugs that the patients had been taking and they should state whether or not they have been administered the monoclonal antibody against INF-1.  In addition, the discussion should be more focused upon their findings and their potential explanations regarding their findings. Finally, apart from improving the discussion, more recent referencing from 2023 and 2024 should be added in the discussion and the reference section. Perhaps the authors might want to comment on the monoclonal antibody against INF-1 as well, which has very recently been released.

Comments on the Quality of English Language POOR QUALITY. NEEDS SUBSTANTIAL IMPROVEMENT.

Dear Reviewer!

The information about treatment added in table 1. Nobody received anifrolumab (antibodies against IFN-1). New references were cited and were added. The discussion elaborated according your recommendations. The English editing was also done.

Dear Reviewer!

I hope the manuscript became better after your suggestions and recommendations.

On behalf of the Authors

Mikhail Kostik, MD, PhD, Professor

Round 2

Reviewer 2 Report

Comments and Suggestions for Authors

Thank you for revising the manuscript in accordance with some of my comments, however, some of them are still to be assessed. 

1. Many abbreviations are still to be spelled out (e.g. CANDLE, SAVI, SPENCD, SLEDAI, ECLAM…). 

2. There are still some typos to revise such as line 126 (37 (were) treated with CFM) and 129 “the half of patients…” and many others.

3. Please use only past tense in results. 

Comments on the Quality of English Language

Fair

Author Response

Dear Reviewer!

Thank you so much for your positive evaluations of our manuscript. Our answers (A) on your queries (Q) are below and highlighted by color in the manuscript. The English editing done.

Q1. Many abbreviations are still to be spelled out (e.g. CANDLE, SAVI, SPENCD, SLEDAI, ECLAM…). 

A1. Dear Reviewer! The full name added the first time in the text abbreviations mentioned (for frequent abbreviations), also we added the full name in the bottom of the table and whole list of abbreviations added at the end of the manuscript. All full names provided in the alphabetical order. Additionally, we created the whole list of the abbreviations at the end of the manuscript.

Q2. There are still some typos to revise such as line 126 (37 (were) treated with CFM) and 129 “the half of patients…” and many others.

A2. The manuscript was checked and multiple typos were fixed.

Q3. Please use only past tense in results. 

A3. Dear Reviewer! Done.

Dear Reviewer!

I hope the manuscript became better after your suggestions and recommendations.

On behalf of the Authors

Mikhail Kostik, MD, PhD, Professor

Reviewer 3 Report

Comments and Suggestions for Authors

Accept in current form

Author Response

Dear Reviewer!

Thank you so much for your postive evaluation

On behalf of the Authors

Mikhail Kostik, MD, Phd, Professor